# Immunomodulatory Effects of IFNα on T and NK Cells in Chronic Myeloid Leukemia Patients in Deep Molecular Response Preparing for Treatment Discontinuation

**DOI:** 10.3390/jcm11195594

**Published:** 2022-09-23

**Authors:** Maria Cristina Puzzolo, Massimo Breccia, Paola Mariglia, Gioia Colafigli, Sara Pepe, Emilia Scalzulli, Elena Mariggiò, Roberto Latagliata, Anna Guarini, Robin Foà

**Affiliations:** 1Hematology, Department of Translational and Precision Medicine, Policlinico Umberto 1, ‘Sapienza’ University, 00161 Rome, Italy; 2Hematology, Department of Molecular Medicine, ‘Sapienza’ University, 00161 Rome, Italy

**Keywords:** chronic myeloid leukemia, interferon-alpha, NK cells, T lymphocytes, treatment discontinuation

## Abstract

A deep and stable molecular response (DMR) is a prerequisite for a successful treatment-free remission (TFR) in chronic myeloid leukemia (CML). In order to better identify and analyze potential candidates of successful TFR, we examined the phenotypic and functional host immune compartment in DMR patients who had received TKI treatment only (TKI-only) or had been previously treated with interferon-alpha (IFNα + TKI) or had received IFNα treatment only (IFNα-only). The T/NK-cell subset distribution, NK- and T-cell cytokine production, activation and maturation markers were measured in 44 patients in DMR treated with IFNα only (9), with IFNα + TKI (11) and with TKI-only (24). IFNα + TKI and TKI-only groups were eligible to TKI discontinuation according to the NCCN and ESMO guidelines (stable MR4 for more than two years). In IFNα-treated patients, we documented an increased number of lymphocytes capable of producing IFNγ and TNFα compared to the TKI-only group. In INFα + TKI patients, the percentage of NKG2C expression and its mean fluorescence intensity were significantly higher compared to the TKI-only group and to the INFα-only group in the CD56dim/CD16+ NK cell subsets (INFα + TKI vs. TKI-only *p* = 0.041, *p* = 0.037; INFα + TKI vs. INFα-only *p* = 0.03, *p* = 0.033, respectively). Furthermore, in INFα-only treated patients, we observed an increase of NKp46 MFI in the CD56bright/CD16- NK cell subset that becomes significant compared to the INFα + TKI group (*p* = 0.008). Our data indicate that a previous exposure to IFNα substantially and persistently modified the immune system of CML patients in memory T lymphocytes, differentiated NKG2C+ “long-lived” NK cells responses, even years after the last IFNα contact.

## 1. Introduction

Chronic myeloid leukemia (CML) is a myeloproliferative disorder characterized by the Philadelphia chromosome and a balanced translocation between the long arms of chromosomes 9 and 22 [1]. The cytogenetic abnormality results in the expression of a constitutively active tyrosine kinase protein called BCR/ABL1, responsible for the disease [2]. In the last 15 years, the outcome of this disease has dramatically improved following the introduction of tyrosine kinase inhibitors (TKIs) as a specific and selective target therapy against BCR/ABL1 activity. A recent analysis of the Swedish registry has shown that the overall survival of CML patients is now similar to that of the general population [3]. A deep and stable molecular response (DMR) is the final endpoint to prevent progression and to enable a possible treatment discontinuation, in order to avoid long-term off-target effects of TKI treatment [4]. Increasing evidence suggests that the host immune compartment might impact the pathogenesis and long-term outcome of CML patients [5,6]. In particular, inhibitory microenvironment pathways are predominant in newly diagnosed patients with an expansion of regulatory T cells (Tregs) and myeloid-derived suppressor cells (MDSCs), and of the expression of programmed death-1 (PD1) inhibitory molecules on CD4+/CD8+ T cells [7].

Many studies have found that T cells with memory phenotypes have an important role in antitumor immunity and in its persistence [8,9].

The longevity of immunological memory T cell subsets has been observed lasting for decades in humans [10,11].

Studies on certain immune cells classified as innate have revealed that natural killer (NK) cells can exhibit memory-like properties to certain viruses and haptens. As explained by Cerwenka and Lanier, NK cells can also remember inflammatory cytokine milieus that imprint long-lasting non-antigen-specific NK cell effector function [12]. The memory properties of NK cells have been documented in mice [13]. In humans, an expansion of differentiated NKG2C+ NK cells in response to CMV infection has already been proven, providing long-term protection [14]. NK cells are reduced at diagnosis and then restored in patients who achieve a major molecular response after treatment with TKIs with a more mature cytolytic phenotype (CD57+ CD62L-) [15]. Following discontinuation of TKIs, NK cells are increased only in patients who maintain the molecular response [16,17]. Relapses after discontinuation occur because quiescent leukemic stem cells (LSCs) are insensitive to TKIs [18].

Unlike TKIs, interferon-alpha (IFNα) targets LSCs, and an anti-leukemic activity in patients achieving long-lasting cytogenetic responses has been demonstrated [19]. Improving the immune surveillance is an attractive possible strategy in order to increase the rate and persistence of treatment-free remissions (TFR).

In an attempt to better identify potential candidates of successful treatment discontinuation, we have examined the phenotypic and functional host immune compartment-T cells, NK and NK cell subsets-of patients in DMR who had received INFα prior to TKI treatment (IFNα + TKI) or only IFNα (IFNα-only) and compared it with that of patients treated only with TKIs (TKI-only). Our results show that, in CML patients, IFNα treatment modulates and potentiates the in vivo host immunologic compartment and paves the way to the optimization of immunotherapeutic strategies aimed at achieving and maintaining a DMR as a possible pre-requisite for TKI discontinuation.

## 2. Materials and Methods

A total of 44 CML patients in DMR eligible to TKI discontinuation according to recent NCCN [20] and ESMO guidelines [21] (stable MR4 for more than two years) was investigated as a cross-sectional study, considering as an inclusion criterion only the achievement of a sustained DMR regardless of the TKI used. Peripheral blood (PB) samples were obtained from 9 patients treated only with IFNα, 11 patients treated with IFNα + TKI and 24 patients treated only with TKI; all experiments were performed before any attempt of TKI discontinuation. Patients included in the study treated only with IFNα had discontinued IFNα for 16.4 years on average, while patients treated with IFNα + TKI included in the study had discontinued IFNα for 14.5 years but continued with TKIs. An informed consent for biologic studies was obtained from patients or their legal guardians in accordance with the Declaration of Helsinki. For each patient, the T/NK-cell subset distribution, NK and T-cell cytokine production, NK activation and maturation markers were studied. Phenotypic analyses were performed on a FACSCanto flow cytometer using the FACSDiva software (BD Biosciences, San Jose, CA, USA). Evaluation of T and NK cells on primary PB samples was performed using different combinations of the following monoclonal antibodies (mAbs): CD3, CD4, CD8, CD16, CD56, CD25, CD69, CD62L, CD57 (BD Biosciences). In particular, for NK activation and maturation markers, NKp30, NKp44, NKp46 and NKG2C activating receptors (R&D System, Minneapolis, MN, USA) were tested. The expression of the NKG2D and DNAM1 activator receptor on NK cells was evaluated using NKG2D (R&D System) or DNAM1 (AbD Serotec, Oxford, UK) unconjugated mAbs, followed by secondary FITC-conjugated IgG1 (BioLegend, San Diego, CA, USA) staining. For cytokine analysis, T and NK cells activated with PMA and ionomycin, were permeabilized and stained with the following mAbs: IFNγ, TNFα, CD3, CD4, CD16, CD56. T cells were also analyzed using the HumanTh1/Th2/Th17 Phenotyping Kit (BD Biosciences) following the manufacturer’s protocol. Cell populations of interest are reported as a proportion of total lymphocytes, derived from side scatter vs. forward scatter gated lymphocytes. Cell surface expression was quantified as the mean fluorescence intensity (MFI) of values obtained with specific mAbs compared with values given by isotype controls.

### 2.1. Assessment of Cytotoxic Activity by NK Cells

The NK cytotoxic activity was quantified by flow cytometry using the NK TESTTM kit (Glycotope, Biotechnology GmbH, Heidelberg, Germany) following the manufacturer’s protocol. Peripheral blood mononuclear cells (PBMC) were obtained by centrifugation of fresh blood on a Ficoll-Histopaque (Axis-Shield, Oslo, Norway) gradient and cryopreserved to study the NK cytotoxic activity against the K562 cell line. The cytotoxicity was performed only if the viability of thawed PBMCs was greater than 90%. Briefly, 10^6^ cells/mL PBMC, were mixed with target cells (labelled K562 cells) and incubated for 120 min. Immediately before the flow cytometric analysis, a DNA staining was carried out. Data were acquired by a FACSCanto I flow cytometer (BD Biosciences). For each analysis, 100,000 events were acquired and analyzed using the FACSDiva software (BD Biosciences). Spontaneous cell death (without effector cells) was considered as control. In this way, the percentage of target cells killed by effector NK cells was determined.

### 2.2. Statistical Analysis

Data were checked for normality with the Kolmogorov-Smirnov test. Due to non-parametric data distribution, comparisons of continuous parameter in different groups were evaluated using the Kruskal-Wallis (KW) non-parametric statistics test. Significance values have been adjusted by Bonferroni correction for multiple tests. *p*-values of 0.05 or less were considered significant and data are given as median percentage frequencies and interquartile range (IR).

## 3. Results

### 3.1. Characteristics of Patients

Overall, 44 patients in DMR were studied. The median age of the whole cohort was 63 years (range 25.6–80.9). Nine of the 44 patients received only IFNα. The median age of this group was 67 years (range 55.9–80.9). In this patient cohort, the median time from diagnosis was 26.9 years (range 19.5–30.3) and 26.7 years from the start of IFNα; the last contact with IFNα was 16.4 years earlier. In these patients, the first DMR was achieved at a median of 16.8 years (range 14.6–27.7). Eleven of the 44 patients received first IFNα and subsequently a TKI. The median age of this group was 59 years (range 50.1–79.5). In this patient cohort, the median time from diagnosis was 17.5 years (range 16.2–28.5) and 17.3 years from the start of IFNα; the last contact with IFNα was 14.5 years earlier. In these patients, the first DMR was achieved at a median of 5.1 years (range 1.2–9). After IFNα resistance and intolerance, all patients received imatinib. Twenty-four patients were treated with a TKI as first-line therapy: the median age was 57 years (range 25.6–78.4). The median time from diagnosis in this cohort was 11.8 years (1.2–25.8) and the median time from the first stable DMR was 3.4 years (range 0–11.6). Seventeen patients received imatinib, 6 patients nilotinib and 1 patient dasatinib as first-line treatment (Table 1).

### 3.2. Phenotypic Characterization of the Three Groups

No differences were observed in the overall percentage of T-lymphocyte populations among the three groups [63.9% (IQR 55.1–77.2%) in the IFNα-only group vs. 57.4% (IQR 32.6–64.7%) in the IFNα + TKI group vs. 60.1% (IQR 44–67.2%) in the TKI-only group, *p* = ns] or in the CD3 + CD4+ fraction [43.2% (IQR 32.2–48.9%) vs. 39.7% (IQR 23.1–47.1%) vs. 41.5% (IQR 24.3–52.2%)], and in the percentage of NKT cells [5.1% (IQR 1.6–7.9%) vs. 3.6% (IQR 2–9.3%) vs. 5.0% (IQR 3.3–8.9%)]. We instead documented a significant increase in the percentage of CD3 + CD8+ in patients who had received only IFNα if compared to the TKI-only group: 23.6% (IQR 19.1–29.1%) vs. 13.6% (IQR 10.6–19.7%) (*p* = 0.029), while the percentage of CD3 + CD8+ in the IFNα + TKI group was 17.7% (IQR 10.6–19.3%). We also evaluated changes in the absolute lymphocyte counts and recorded a significantly increased number of T cells in patients treated only with INFα compared to the IFNα + TKI (*p* = 0.024) and TKI-only groups (*p* = 0.020) [1425/µL (IQR 1161/µL–1739/µL) in the IFNα-only group vs. 981/µL (IQR 528/µL–1313/µL) in the IFNα + TKI group vs. 977/µL (IQR 610/µL–1280/µL) in the TKI-only group]. In line with the percentage data, a significant increase in the absolute number of CD3 + CD8+ T cells was observed in the IFNα-only group compared to the IFNα + TKI group (*p* = 0.039) and the TKI-only group (*p* = 0.015) [519/µL (IQR 340/µL–664/µL) vs. 331/µL (IQR 138/µL–391/µL) vs. 238/µL (IQR 154/µL–390/µL)]. The percentage and absolute count of NK cells was similar between the three patients’ groups and also a further characterization of NK cell subpopulation showed no differences in CD56brightCD16-, CD56brightCD16dim and CD56dimCD16+ (Appendix A).

### 3.3. Cytokine Production

We detected an increased number of lymphocytes capable of producing IFNγ and TNFα in patients treated with INFα (both IFNα-only and INFα + TKI groups) compared to the TKI-only group. In particular, analyzing the percentage of IFNγ production by lymphocyte populations, we observed a significant increase of CD4 + IFNγ+ T cells in the IFNα + TKI group compared to the TKI-only group (*p* = 0.035). A slight but not significant increase of CD8 + IFNγ+ T cells in the two groups of INFα-treated patients compared to the TKI-only group and a significant increase of NKT cells in the IFNα-only group vs. TKI-only treated patients (*p* = 0.011) were also observed (Figure 1, Appendix A).

The percentage of CD3 + CD4 + IFNγ+ T cells (Th1) was also analyzed using the HumanTh1/Th2/Th17 Phenotyping Kit that confirmed a significant increase in the INFα + TKI group compared to the TKI-only group (*p* = 0.013) (Figure 2, Appendix A). A significant increase was also detected in CD3 + CD4 + IL-4+ T cells (Th2) cells in the IFNα-only group compared to the TKI-only group (*p* = 0.011). No statistical differences were observed in T-helper cytokine production for CD3 + CD4 + IL-17A+ T cells (Th17) (Figure 2).

The production of IFNγ did not differ in the whole NK population and among the NK subpopulations (CD56bright/CD16-, CD56dim/CD16+) (Figure 1, Appendix A).

For each analysis, 100,000 events were acquired and analyzed using the FACSDiva software (BD). In particular, lymphocytes were initially gated according to forward and side scatter features. An additional gate was established for CD4+ cells. The Th1, Th2 and Th17 cell populations were reported as percentages of the CD4+ lymphocyte population. The absolute number of lymphocyte subpopulations was calculated by multiplying the immunophenotype percentages by the absolute number of lymphocytes obtained from the full blood count of the same sample. In line with the data obtained analyzing the percentages of cytokine producing T cells, an increase in the absolute number of Th1 and Th2 cells was recorded in IFNα-only group, with a significant difference between the IFNα-only group and the TKI-only group (*p* = 0.034; *p* = 0.003, respectively) (Appendix A).

We documented in the IFNα-only patients a significant TNFα increase in the CD56bright/CD16- NK cell subpopulation (IFNα-only vs. INFα + TKI *p* = 0.010; IFNα-only vs. TKI-only *p* = 0.04) (Figure 3, Appendix A). Along this line, an increase in the absolute count of TNFα producing T cells was confirmed in the INFα-only group compared to TKI-only patients; in particular, a significant increase in CD8+ T cells was found (*p* = 0.04) (Appendix A).

### 3.4. Maturation Stage, Activation Markers and Activating Receptors of NK Subpopulation

In INFα + TKI patients, the percentage of NKG2C expressing NK cells and its mean fluorescence intensity (MFI) were significantly higher compared to the TKI-only group and to the INFα-only group in the CD56dim/CD16+ NK cell subsets (Figure 4). In particular, the percentage of cells expressing NKG2C in the CD56dim/CD16+ NK cell subset was 4.5% (IQR 2.6–14.9%) in the IFNα-only group, 26.7% (IQR 11.2–37.5%) in the IFNα + TKI group and 6.2% (IQR 2.9–10.3%) in the TKI-only group (INFα + TKI vs. TKI-only *p* = 0.041; INFα + TKI vs. INFα-only *p* = 0.03). Furthermore, the NKG2C MFI in the CD56dim/CD16+ NK cell subset was 958 (IQR 714–1710) in the IFNα-only group, 1958 (IQR 1523–2382) in the IFNα + TKI group and 1170 (IQR 1039–1653) in the TKI-only group (INFα + TKI vs. TKI-only *p* = 0.037; INFα + TKI vs. INFα-only *p* = 0.033). We found no differences with regard to the DNAM-1, NKG2D, NKp30, NKp44 expression among the three groups, both in terms of percentage and MFI (Appendix A). Furthermore, in IFNα-only treated patients, we observed a significant increase of NKp46 MFI in the CD56bright/CD16- NK cell subset when compared to the IFNα + TKI group. It was in fact 6171 (IQR 2654–10,144) in the IFNα-only group, 1050 (IQR 525–3486) in the IFNα + TKI group and 1666 (IQR 1093–3664) in the TKI-only group (INFα-only vs. INFα + TKI; *p* = 0.008).

### 3.5. Cytolytic Activity of NK Cells

We also carried out a quantitative determination of the cytotoxic activity of NK effector cells against pre-stained K562 target cells in a total of 27 patients. Six of the 27 patients belonged to the IFNα-only group, 7 to the IFNα + TKI group and 14 to the TKI-only group. Analyzing the NK effector: target (E:T) cell ratio trend from 100:1 to 12.5:1, we observed that effector cells derived from the IFNα-only and IFNα + TKI group cells always showed a higher cytotoxic activity compared to that of the TKI-only group (Figure 5, Appendix A). Focusing on the 50:1 E:T ratio against the K562 cell line, effector cells from the INFα + TKI group of patients showed a higher NK activity than the two other groups: 42.9% (IQR 20.5–60.0%) in the IFNα + TKI group compared to 37.7% (IQR 20.2–56.8%) in the IFNα-only group and 33.9% (IQR 10.6–51.0%) in the TKI-only group.

### 3.6. Preliminary Data about Discontinuation Outcome

Five of the 11 INFα + TKI group patients (45.4%) discontinued treatment and 1 (20%) has relapsed after a median time of 3 months; 11 of the 24 TKI-only group patients (45.8%) discontinued treatment and 5 (45.5%) have relapsed after a median time of four months. Patients treated with INFα + TKI had discontinued TKI after a median time of treatment of five years, while the TKI-only group patients had discontinued TKI after a median of four years. All patients restarted with the same TKI (imatinib in four patients and nilotinib in two patients) and regained a MR4 after a median time of five months. None of them has so far progressed.

A correlation between the T/NK-cell subset distribution, the modulation of activation and maturation markers and cytokine production, and outcome after TKI discontinuation was carried out; no significant differences were observed in the IFNα + TKI group of patients, possibly due to the low number of patients belonging to specific subgroups. Instead, differences were observed analyzing all 16 patients who discontinued TKI treatment (5 within the INFα + TKI group and 11 within the TKI-only group). Overall, the data highlighted in patients remaining in TFR show a significant increase in the percentage of NK cells (*p* = 0.042), in particular in the CD56dim/CD16+ subpopulation (*p* = 0.042), and in the percentage of activating receptors NKp44 on the CD56bright/CD16- (*p* = 0.008) and CD56brigh/tCD16dim (*p* = 0.005) subpopulations compared to patients who relapsed after TKI discontinuation. A significant increase of the NKp44 receptor was also confirmed in terms of absolute counts on total NK cells (*p* = 0.031) and in the CD56bright/CD16- (*p* = 0.008) and CD56brigh/tCD16dim (*p* = 0.022) subpopulations. In addition, we detected an increased number of CD56bright/CD16- cells capable of producing IFNγ (*p* = 0.042).

## 4. Discussion

Several pieces of evidence point to a role of the immune system in CML, such as the therapeutic effect of allogeneic stem cell transplant or the activity of donor lymphocyte infusions (DLI) post-transplant [22]. The recent introduction of treatment discontinuation (TFR) as a final endpoint in the management of CML patients has highlighted the possible impact of the immune system in controlling residual disease. Patients with increased NK cell counts prior to treatment discontinuation have been shown to have a reduced relapse rate as demonstrated in the STIM1 analysis and in the EUROSKI trial [23,24]. Furthermore, Hughes et al. [25] reported that CML patients who achieved a DMR after TKI therapy had a more mature cytolytic CD57+ NK cell phenotype, with a normal restoration of this compartment, increased effector cytotoxic T lymphocytes capable of inducing an immune response against leukemia associated antigens and reduced immune myeloid derived suppressor cells and Tregs.

The results of our study confirm and extend the above reported results, showing that, in patients previously treated with IFN, an increased number of lymphocytes capable of producing IFNγ and TNFα were detected compared to the TKI-only group. In addition, a previous in vivo exposure to IFNα was capable of inducing a persistent modification of the host immune system not only with an enhancement of memory T lymphocytes, but also of differentiated NKG2C+ “long-lived” NK cells, an increased NKp46 MFI on the CD56bright/CD16- NK cell subset and a modulation of the NK cell cytolytic activity, even after a long time period from the last exposure to IFNα.

Earlier studies have shown that NK cells can persist over time and can respond to the same stimulus for a second time [26,27]. The existence of memory NK cells in humans is suggested by the expansion and persistence of NKG2C+ NK cells following cytomegalovirus infection [28]. Furthermore, a mature NK-cell phenotype has been shown in non-relapsing patients [17].

Among the natural cytotoxicity receptor (NCR) family, NKp46 is a major NK cell-activating receptor involved in the recognition of cancer, bacterial and virus-infected cells [29]. The NKp46 receptor expressed by all human NK cells is involved in natural cytotoxicity mediated by freshly (NKp46brigh) derived NK cells [30]. Importantly, it has been shown that NKp46 plays an important role in the recognition and prevention of tumor metastasis [31].

Our observation that in CML patients in DMR previously exposed to IFNα NK cells are not increased in percentage but are more active suggests that IFNα stimulates the host immune compartment against residual disease. Examining the percentage of Th1, Th2 and Th17 cytokine producing lymphocyte cell subsets, we indeed observed an improvement of the host immune surveillance documented by a significant increase in Th1 cytokines (i.e., IFNγ) and a decrease, though not significant, in Th2 cytokines (i.e., IL-4) in the INFα + TKI group. No differences were observed in Th17 cytokines (i.e., IL-17).

These data provide the basis for planning therapeutic strategies aimed at increasing the rate of patients who can successfully discontinue TKI treatment and further point to the existence of leukemia-specific responses in CML. The first evidence was by Kolb et al. [32] who showed that allogenic DLIs could produce durable remissions in about 75% of CML patients who relapsed after a transplant. Indeed, the first evidence that IFNα may mediate CML immunity derived from a study by Molldrem et al. [33] who observed an association between the presence of high-avidity proteinase-3 specific cytotoxic T lymphocytes and the achievement of a cytogenetic response in HLA-A 0201-positive individuals. There are several possible mechanisms induced by IFNα. The first is based on a direct antigen-presenting cell-independent activation of T cells [34,35] with a prolonged survival of activated cells (cytotoxic T cells do not undergo apoptosis in vivo and mediate anti-leukemic function). In addition, IFNα promotes dendritic cell maturation with an upregulation of MHC class I and II molecules needed to present antigens to effector T cells [36,37]. Another mechanism involves CD14-positive monocytes that have shown an increased transcription of PR3, capable of promoting self-antigen presentation in antigen-presenting cells [38]. In vivo, Burchert et al. [39] showed the effects of IFNα maintenance after the combination of IFNα and imatinib. Relapse-free survival was described to be 73% after a median follow-up of 7.9 years and after a median time of 2.8 years 9 out of 20 patients remained in treatment-free remission in DMR [39]. The interest on targeted therapies and the role of the host immune compartment in Ph+ leukemias is extending beyond CML. Our group has recently demonstrated that, in adult Ph+ acute lymphoblastic leukemia, high rates of molecular responses can be obtained by a combined targeted and immunotherapeutic strategy as induction/consolidation without systemic chemotherapy [40]. A marked in vivo modulation of the host immune compartment, documented by a reduction of T-regulatory cells and a marked proliferation of immunocompetent T, T-NK and NK cells, could also be shown [40,41]. In conclusion, the combination of currently available TKIs with immunotherapeutic strategies may have a role in achieving and maintaining a DMR as a possible pre-requisite for TKI discontinuation in patients with CML. Clinical prospective trials should be defined to test the effect of immune-strategies plus TKIs in the long-term not only in preparation for a possible treatment-free remission but also after initial molecular recurrence after discontinuation.

## Figures and Tables

**Figure 1 jcm-11-05594-f001:**
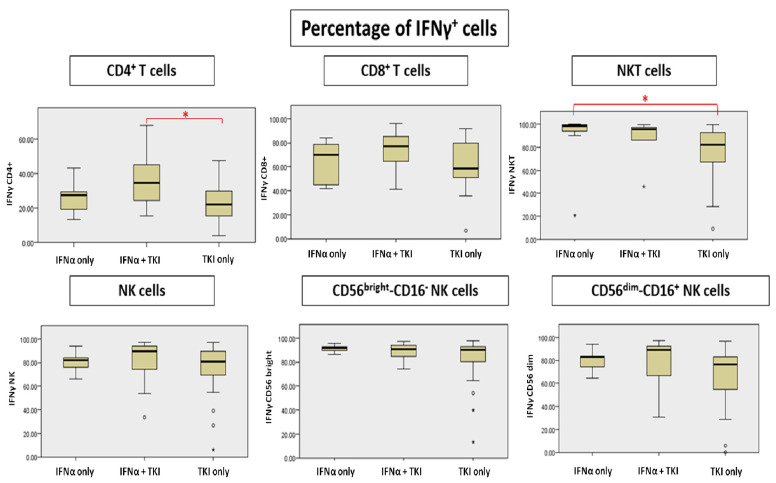
Differences in percentage of IFNγ(Interferon-gamma)-producing cells. * Significant differences are marked with red.

**Figure 2 jcm-11-05594-f002:**
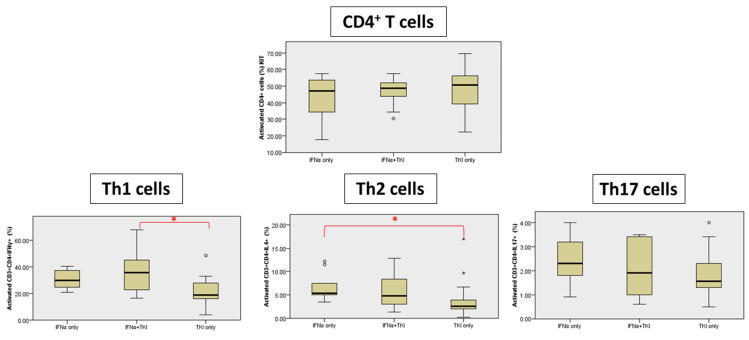
T-cell cytokine production (expressed as percentage of cytokines produced by Th1, Th2 and Th17 cells). * Significant differences are marked with red.

**Figure 3 jcm-11-05594-f003:**
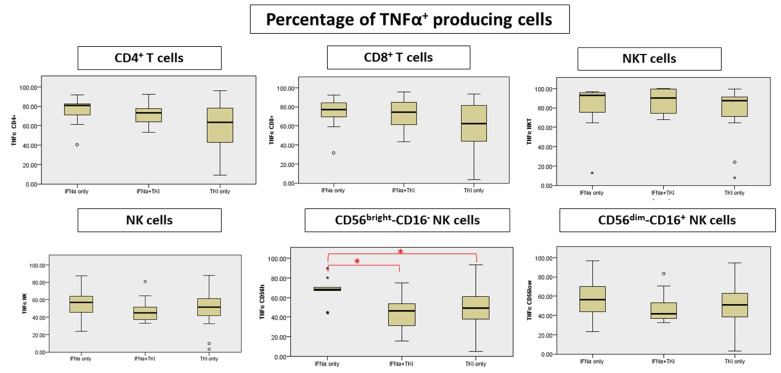
Differences in percentage of TNFα-producing cells. * Significant differences are marked with red.

**Figure 4 jcm-11-05594-f004:**
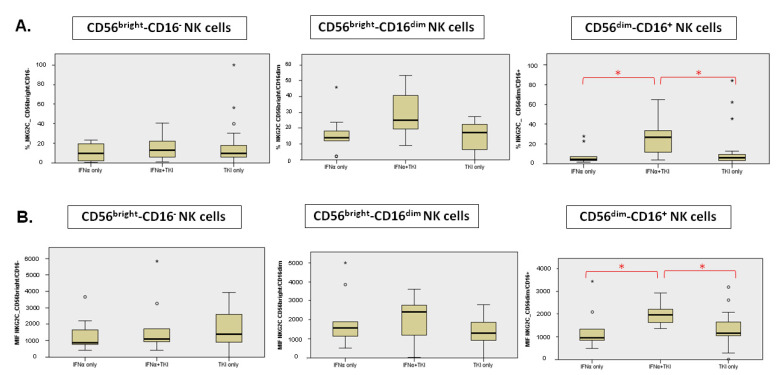
Differences of NKG2C expression in percentage (**A**) and MFI (**B**). * Significant differences are marked with red.

**Figure 5 jcm-11-05594-f005:**
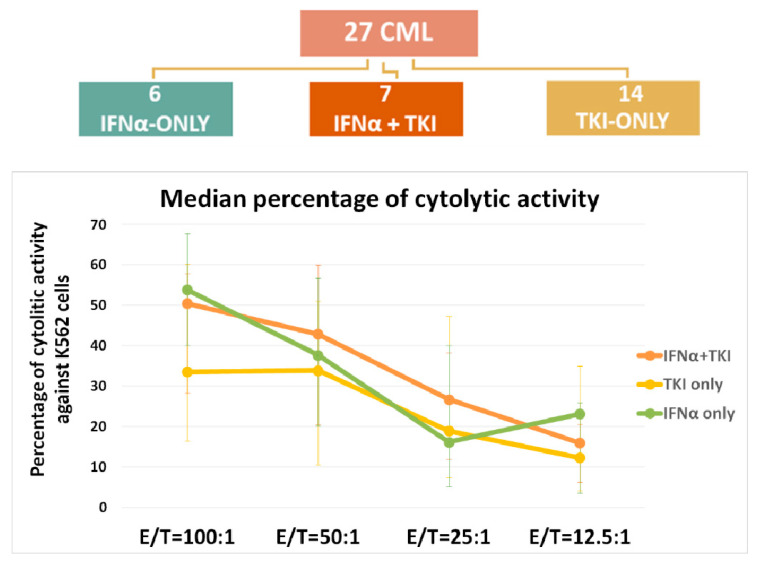
NK test in the three groups. Effector: Target (E:T) cell ratio ranged from 100:1 to 12.5:1.

**Table 1 jcm-11-05594-t001:** Characteristics of patients.

	Total No.	INFα-Only	INFα + TKI	TKI-Only
No. of patients	44	9	11	24
Age (years)	63 (25.6–80.9)	67 (55.9–80.9)	59 (50.1–79.5)	57 (25.6–78.4)
Gender (M/F)	23/21	6/3	4/7	13/11
Sokal score:				
Low	37	8	10	19
Intermediate	5	1	1	3
High	2	0	0	2
Years from diagnosis	14.7 (1.2–30.3)	26.9 (19.5–30.3)	17.5 (16.2–28.5)	11.8 (1.2–25.8)
Years from first IFNα treatment	21.4 (16.1–29.9)	26.7 (19.3–29.9)	17.3 (16.1–27.3)	
Years from last IFNα treatment	14.5 (11–28.7)	16.4 (11.6–28.7)	14.5 (11–16)	
Years of IFNα treatment	4.7 (0.9–13.9)	6.7 (1.1–13.9)	3.1 (0.9–13.40)	
Years from first TKI treatment	13.4 (1.2–17.1)		14.3 (11–17.1)	11.8 (1.2–14.9)
Years from first DMR	5.1 (0–27.7)	16.8 (14.6–27.7)	5.1 (1.2–9)	3.4 (0–11.6)
Type of TKI:				
Imatinib			11	17
Nilotinib			-	6
Dasatinib			-	1

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
