# Peer review of "Immunomodulatory Effects of IFNα on T and NK Cells in Chronic Myeloid Leukemia Patients in Deep Molecular Response Preparing for Treatment Discontinuation"

_jcm, 2022, doi:10.3390/jcm11195594_

Round 1

Reviewer 1 Report

Hypothesis:

Authors have suggested the  deep and stable molecular response (DMR) is a prerequisite for a successful treatment-free remission (TFR) in chronic myeloid leukemia (CML). 

Aim:

In order to better identify and analyse potential candidates of successful TFR a  total of 44 patients were examined

Inclusion Criteria:

IFNα+TKI and TKI-only groups were eligible to TKI discontinuation according  to the NCCN and ESMO guidelines (stable MR4 for more than 2 years) 

Patient Selection:

the phenotypic and functional host  immune compartment in DMR patients was performed in Peripheral blood (PB) samples

·      had received TKI treatment only (TKI-only) - 24 patients- discontinuation according  to the NCCN and ESMO guidelines (stable MR4 for more than 2 years) 

·      had been previously treated with interferon-alpha (IFNα+TKI) – 11 patients- had discontinued IFNα for 14.5 years but still continued with TKIs.

·      had received IFNα treatment only  (IFNα-only)- 9 patients- had discontinued IFNα for 16.4 years on average

Markers assessed :

The T/NK-cell subset distribution- using different combinations of the following monoclonal antibodies (mAbs): CD3, CD4, CD8, CD16, CD56, CD25, CD69, CD62L, CD57 (BD Biosciences)

NK- and T-cell activation and maturation markers - NK activation and maturation markers, NKp30, NKp44, NKp46 and NKG2C activating receptors

For cytokine analysis, T and NK cells activated with PMA and ionomycin, were permeabilized and stained with the following mAbs: IFNγ, TNFα, CD3, CD4, CD16, CD56. 

T cells were also analysed using the HumanTh1/Th2/Th17 Phenotyping Kit (BD Biosciences) following the manufacturer’s protocol.

Results:

·      IFNα-treated patients compared to the TKI-only group 

a.     showed an increased number of lymphocytes capable of producing IFNγ (TH2 cytokine) and TNFα in IFNα-treated patients- suggesting that IFNα stimulates the host immune compartment against residual disease.

·      In INFα+TKI patients compared to the TKI-only group, 

a.     the percentage of NKG2C expression and its mean fluorescence intensity were significantly higher in INFα+TKI patients and suggested that differentiated NKG2C+ “long-lived” NK cells, an increased NKp46 MFI on the 294 CD56bright/CD16- NK cell subset and a modulation of the NK cell cytolytic activity, even 295 after a long time period from the last exposure to IFNα.

·      In the INFα-only group in the CD56dim/CD16+ NK cell subsets 

a.     INFα+TKI vs TKI-only p=0.041, 30 p=0.037

b.     INFα+TKI vs INFα-only p=0.03, p=0.033. 

·      In INFα-only treated patients we observed an increase of NKp46 MFI in the CD56bright/CD16- NK cell subset 

a.     INFα+TKI group vs INFα-only (p=0.008)-corroborated with previous reports that NKp46 plays an important role in the recognition and prevention of 306 tumor metastasis

Inference:

Authors suggest 

·      previous exposure to IFNα substantially and persistently modified the immune system of CML patients in memory T lymphocytes, differentiated NKG2C+ “long-lived” NK cells responses, even years after the last IFNα contact.

Comments:

In Abstract authors have reported that “ previously treated with interferon-alpha (IFNα+TKI) – 11 patients- had discontinued IFNα for 14.5 years but still continued with TKIs.

I think still continued with TKIs will affect the overall results. Contrarily in “3.6 Preliminary data about discontinuation outcome “ authors have stated that “Patients treated with INFα+TKI, had discontinued TKI for 5 years on average”. Kindly explain the discrepancy.

In Introduction section authors must include more about the mechanism of memory T lymphocytes, differentiated NKG2C+ “long-lived” NK cells responses.

Similarly the explanation of results in the discussion section is not up to the mark.

More experiments should be carried out to validate the hypothesis.

Author Response

Dear Editor,

Please find enclosed our revised paper “IMMUNOMODULATORY EFFECTS OF IFNα ON T AND NK CELLS IN CHRONIC MYELOID LEUKEMIA PATIENTS IN DEEP MOLECULAR RESPONSE PREPARING FOR TREATMENT DISCONTINUATION” (jcm-1885911), which has been revised according to the referees’ comments and suggestions. We have detailed below all changes made.

Best regards.

Dr. Maria Cristina Puzzolo

Reviewer 1

  1. As suggested, we clarify the discrepancy between the abstract and the section 3.6
  2. In the introduction section, we include a new paragraph about the mechanism of memory T lymphocytes, differentiated NKG2C+ “long-lived” NK cells responses.
  3. In the results section we detailed the explanation of results related to different experiments

Please see the revised article attached.

Reviewer 2 Report

This study has shown that previous exposure to IFNa has a long term effect on the immune system of CML patients.   

Overall it is a well done study.

Significance of any of the clinical features in Table 1 should be shown; for example is the 10 years age difference between INFa-only and TKI-only significant could it impact on the findings?  Or indeed the time from 1st DMR?  

The axis titles on Figure 1 are very small and difficult to read even at 300% zoom.  

What is the difference between CD4+ T cells in Figure 1 and that in Figure 2?

Error bars should be shown for Figure 5 and indication if the any of the ratios are significantly different.

It would be useful for the authors to suggest how the combination of TKI with immunotherapy strategies might be done or monitored as an indication of discontinuation or even restarting TKI

Author Response

Dear Editor,

Please find enclosed our revised paper “IMMUNOMODULATORY EFFECTS OF IFNα ON T AND NK CELLS IN CHRONIC MYELOID LEUKEMIA PATIENTS IN DEEP MOLECULAR RESPONSE PREPARING FOR TREATMENT DISCONTINUATION” (jcm-1885911), which has been revised according to the referees’ comments and suggestions. We have detailed below all changes made.

Best regards.

Dr. Maria Cristina Puzzolo

Reviewer 2

  1. Considering the concepts of the study and the relative low number of patients included, due to the specific experiments done for each patient, a p value was not indicative of the messages relative to the study. Indeed, the important messages were included in the results section as stated also by the reviewer 2 that we thank for the judgement.
  2. The axis title was changed as suggested
  3. In figure 1 we shown intracytoplasmic detection of IFNγ in the various lymphocyte populations, including CD4 T-lymphocytes. In figure 2, we focused on cytokines produced by CD4+T cells, in particular produced by Th1, Th2 and Th17 cells  that were analyzed using the HumanTh1/Th2/Th17 Phenotyping Kit (BD Biosciences). The percentage of CD3+CD4+IFNγ+ T cells (Th1) in figure 2, confirmed the significant increase in the INFα+TKI group compared to the TKI-only group already shown in figure 1.
  4. Error bars were included in figure 5
  5. In the conclusions section we stated how the combination TKI+immune therapies should be implemented.

Please see the revised article attached.
